Coral reefs; land-based pollution; water-borne disease; systems health; WASH; pathogens

**Corresponding author:**
Ama Wakwella;
Email: a.wakwella@uq.edu.au

# Integrated watershed management solutions for healthy coastal ecosystems and people

Ama Wakwella[1,2] (iD), Amelia Wenger[1,2,3] (iD), Aaron Jenkins[4,5] (iD), Joleah Lamb[6] (iD), Caitlin D. Kuempel[7,8] (iD), Danielle Claar[9], Chris Corbin[10], Kim Falinski[11], Antonella Rivera[12] (iD), Hedley S. Grantham[13,14] (iD) and Stacy D. Jupiter[15] (iD)

[1]School of Earth and Environmental Sciences, University of Queensland, St Lucia, QLD, Australia; [2]Centre for Biodiversity and Conservation Science, The University of Queensland, St Lucia, QLD, Australia; [3]Global Marine Program, Wildlife Conservation Society, Bronx, NY, USA; [4]Sydney Institute for Infectious Diseases, Sydney School of Public Health, The University of Sydney, Camperdown, NSW, Australia; [5]Centre for People Place and Planet, School of Science, Edith Cowan University, Joondalup, WA, Australia; [6]Department of Ecology and Evolutionary Biology, University of California – Irvine, Irvine, CA, USA; [7]Australian Research Council Centre of Excellence for Coral Reef Studies, University of Queensland, St Lucia, QLD, Australia; [8]School of Environment and Science, Griffith University, Nathan, QLD, Australia; [9]Nearshore Habitat Program, Washington State Department of Natural Resources, Olympia, WA, USA; [10]United Nations Environment Programme, Cartagena Convention Secretariat, Kingston, Jamaica; [11]Hawai'i and Palmyra Chapter, The Nature Conservancy, Honolulu, HI, USA; [12]Western Caribbean Department, The Coral Reef Alliance, San Francisco, CA, USA; [13]Science and Conservation, Bush Heritage Australia, Melbourne, VIC, Australia; [14]Centre for Ecosystem Science, University of New South Wales, Sydney, NSW, Australia and [15]Melanesia Program, Wildlife Conservation Society, Suva, Fiji

## Abstract

Tropical coastal ecosystems are in decline worldwide due to an increasing suite of human activities, which threaten the biodiversity and human wellbeing that these ecosystems support. One of the major drivers of decline is poor water quality from land-based activities. This review summarises the evidence of impacts to coastal ecosystems, particularly coral reefs, from sediments, nutrients, chemicals and pathogens entering coastal zones through surface and groundwater. We also assess how these pollutants affect the health of coastal human populations through: (1) enhanced transmission of infectious diseases; (2) reduced food availability and nutritional deficit from decline of fisheries associated with degraded habitat; and (3) food poisoning from consumption of contaminated seafood. We use this information to identify opportunities for holistic approaches to integrated watershed management (IWM) that target overlapping drivers of ill-health in downstream coastal ecosystems and people. We demonstrate that appropriate management requires taking a multi-sector, systems approach that accounts for socio-ecological feedbacks, with collaboration required across environmental, agricultural, public health, and water, sanitation and hygiene sectors, as well as across the land–sea interface. Finally, we provide recommendations of key actions for IWM that can help achieve multiple sustainable development goals for both nature and people on coasts.

## Impact Statements

The pollution of water and waterways from land-based human activities has extensive impacts on both human and ecosystem health, contributing to significant global health burdens and loss of critical ecosystem services. Management of pollution is therefore a major focus of multiple sustainable development goals (SDGs) to achieve targets for: zero hunger (SDG 2); good health and well-being (SDG 3); clean water and sanitation (SDG 6); climate action (SDG 13); life below water (SDG 14); and life on land (SDG 15). Despite extensive and complex impacts of poor water quality, pollution control has been highly sectorised and under-resourced, with poor coordination of implementation, often across insufficient scales to realise benefits. This review provides a novel summary of the overlapping impacts of water pollution to downstream public and coastal ecosystem health to support planning and decision-making that benefits a wide range of stakeholders from government, civil society and the private sector. We provide evidence-based suggestions to optimise investments in holistic, integrated watershed management (IWM) to improve water quality and achieve overall systems health, which also provides co-benefits for biodiversity and climate. We also identify the key enabling factors required to coordinate and monitor IWM implementation to achieve desired outcomes. Specifically, the summary of pollution impacts and suggested management strategies provided in this review aim to provide awareness and tools to alleviate impacts to nutrition, water-related disease burdens and food poisoning that arise from poor water quality, which cause devastating economic and health costs disproportionately borne by the poorest countries.





## Introduction

Tropical coastal ecosystems support some of the most diverse and productive environments on Earth and provide millions of people with vital ecosystem goods and services, such as food, livelihoods and coastal protection (Moberg and Folke, 1999; Cesar et al., 2003). However, with over 1.3 billion people in the tropics living within 100 km of coastlines (Sale et al., 2014), coastal ecosystems are becoming increasingly threatened by a suite of local, regional and global human activities, many of which affect water quality (Bellwood et al., 2004; Lotze et al., 2006; Orth et al., 2006). Declining water quality is a primary driver of coastal ecosystem degradation (Crain et al., 2009). Declines in water quality are driven mainly by pollutants from upstream human activities within watersheds flowing into coastal environments and are expected to worsen with increased coastal development and future climate change (Rabalais et al., 2009; He and Silliman, 2019).

Watershed management has received increasing focus as a tool for preserving the health of downstream coastal ecosystems, with research demonstrating critical land–sea linkages for coastal ecosystem health (Carlson et al., 2019; Sahavacharin et al., 2022). Despite the extensive literature and examples of decline, there are few examples of watershed management producing improvements to tropical coastal ecosystem conditions (Wear, 2016). Challenges in achieving measurable success are largely due to the large spatial scale over which interventions often need to be applied within watersheds to adequately address multiple sources of pollution, capacity shortfalls for necessary monitoring, and the temporal lags to detect any changes in water quality and/or ecosystem health within coastal environments (Meals et al., 2010).

Watershed condition also regulates a suite of processes that affect human health and wellbeing, including water filtration, flood management, and the provision of important cultural and recreational services (Jenkins et al., 2018a). Polluted water flowing within watersheds onto coastal environments is a major contributor to global human disease burdens, with poor water quality conservatively estimated to result annually in 1.4 million deaths, 3 million disability-adjusted life years and 12 billion USD in economic losses, a cost disproportionately borne by the poorest countries (Shuval, 2003; Fuller et al., 2022). Yet the influence of watershed management on human health is rarely considered and is largely absent from public health literature (Bunch et al., 2014).

Identifying the overlapping upstream drivers of poor water quality that also create significant risks to public health presents an opportunity to motivate action and leverage long-term and large-scale investments while simultaneously improving coastal ecosystem water quality. By facilitating both human and ecosystem health, watersheds can serve as a focal area for place-based management interventions that serve to promote overall systems health (Cadham et al., 2005; Parkes and Horwitz, 2009; Jenkins et al., 2018b; Jordan and Benson, 2020). Here, we consider systems health as the emergent result of functioning interdependencies, interactions and feedbacks between ecological and socio-cultural settings, behaviour, and physiology, nested across micro-level (e.g., communities of microbes), meso-level (e.g., watersheds) and macro-level (e.g., global climate patterns) domains.

This review aims to: (1) synthesise and summarise the latest science regarding water quality impacts on coastal ecosystems (focused primarily on coral reefs); (2) identify pathways to improve systems health through policy implementation and direct management actions; and (3) provide evidence-based suggestions for strategic investments in watershed interventions across sectors that can help achieve multiple sustainable development goals (SDGs) and other global commitments and targets relating to biodiversity, marine pollution and public health.

## Water quality impacts on coastal ecosystems

The quantity and quality of land-based runoff flowing into adjacent coastal ecosystems is determined by the characteristics of the watershed, such as geology, rainfall, soil type, land cover/vegetation (type and quantity) and slope (Douglas, 1967). There is a large body of evidence that demonstrates how human activities within watersheds alter runoff by removing native vegetation, changing the hydrology, altering microbial communities and adding/increasing pollutants within runoff (e.g., Peters and Meybeck, 2000; Liao et al., 2020).

Several broad pollutant categories are used to describe the pollutants reaching coastal waters from land-based activities. Here, we focus on the following common categories applicable to both human and coastal ecosystem health: sediments, nutrients, persistent organic pollutants (POPs), plastics and microdebris, pathogens, heavy metals, and pharmaceuticals and personal care products (Todd et al., 2010; World Health Organization (WHO), 2016; Kroon et al., 2020). Terrestrially derived sediments, heavy metals and nutrients are naturally transported from soils into coastal environments by ground and surface water, but due to large-scale human activities such as land-clearing (Table 1), the sources and transport into coastal waters has increased drastically, threatening over 30% of coral reefs globally (Andrello et al., 2021). POPs are synthetic organic chemicals that can persist in soils and water and bioaccumulate in organisms. POPs are widely produced across industries (Table 1) both intentionally, such as some insecticides, and unintentionally as by-products, such as dioxins (Weber et al., 2011). Other synthetic pollutants include the nonorganic plastics and microdebris, which can flow into coastal waters from numerous human sources (Table 1) such as trash, litter and weathering of materials like tires (Smith et al., 2018; Macleod et al., 2021). Pathogens are disease-causing microbes and can naturally exist in coastal water and organisms but can also be introduced from land-based sources such as sewage (Table 1). Pharmaceuticals include chemicals used for personal, agricultural or animal health, such as antibiotics, while personal care products include chemicals generally used for cosmetic reasons, such as shampoos and moisturisers (Boxall et al., 2012).

The primary land-based activities creating these pollutants and driving global declines in coastal water quality are land clearing, poor food production practices, urban development, mining and poor wastewater management (domestic and industrial) (Lu et al., 2018). These human activities erode or release pollutants such as sediment, metals, pathogens and nutrients into surface and groundwater, which are then transported downstream to coastal environments (Crain et al., 2009; Amato et al., 2016). The flow of impacts from human activities within watersheds to coastal ecosystems is summarised below (Figure 1).

As outlined in Table 1, pollutants can have multiple sources that can make it difficult to pinpoint which activity in a watershed is having the greatest impact on coastal ecosystems. For example, nutrients and sediments can originate from both wastewater pollution and agricultural runoff (Figure 1). Similarly, pharmaceuticals and personal care products can originate from cosmetics and medications used domestically as well as from medications used

**Table 1.** Key references documenting global/regional linkages between human activities within watersheds and elevated levels of pollutants in runoff to coastal ecosystems

| Human watershed activity | Pollutant | Key references |
|---|---|---|
| Agriculture | <ul><li>Sediments</li><li>Nutrients</li><li>Persistent organic pollutants (e.g., organophosphates and organochlorides in pesticides)</li><li>Heavy metals (e.g., copper in fertilisers, mercury in fungicides)</li><li>Pharmaceuticals (e.g., antibiotics)</li><li>Plastics and microdebris</li></ul> | van Dam et al., 2011; Thorburn et al., 2013; Kroon et al., 2014; MacLeod et al., 2021 |
| Livestock and invasive ungulates | <ul><li>Sediments</li><li>Nutrients</li><li>Pathogens (e.g., zoonotic virus/bacteria)</li><li>Heavy metals (e.g., copper from livestock feed)</li><li>Pharmaceuticals (e.g., antibiotics)</li><li>Plastics and microdebris</li></ul> | Agouridis et al., 2005; McDowell and Wilcock, 2008; Todd et al., 2010; Bartley et al., 2014 |
| Aquaculture | <ul><li>Nutrients</li><li>Persistent organic pollutants (e.g., organotin in molluscicides)</li><li>Heavy metals (e.g., copper in algaecides)</li><li>Pharmaceuticals (e.g., antibiotics)</li><li>Plastics and microdebris</li></ul> | Gräslund and Bengtsson, 2001; Primavera, 2006; Lusher et al., 2017; Wang et al., 2020 |
| Deforestation and burning | <ul><li>Sediments</li><li>Nutrients</li><li>Persistent organic pollutants (e.g., polycyclic aromatic hydrocarbons from burning)</li></ul> | Sundarambal et al., 2010; Todd et al., 2010; Suárez-Castro et al., 2021 |
| Urban development (surface hardening and channel modification) | <ul><li>Sediments</li></ul> | Freeman et al., 2007; Kroon et al., 2014; McGrane, 2016 |
| Mining (including gravel extraction) | <ul><li>Sediments</li><li>Nutrients</li><li>Persistent organic pollutants (e.g., polycyclic aromatic hydrocarbons from coal mining)</li><li>Heavy metals (e.g., lead, nickel, mercury)</li></ul> | Kondolf, 1994; Ahrens and Morrisey, 2005; Todd et al., 2010; van Dam et al., 2011; Shumway, 2020 |
| Wastewater (sewage, domestic, industrial, and storm water) | <ul><li>Sediments</li><li>Nutrients</li><li>Pathogens (e.g., water-associated bacteria/virus)</li><li>Persistent organic pollutants (e.g., oil hydrocarbons from urban storm water)</li><li>Heavy metals (e.g., tin from industrial wastewater)</li><li>Pharmaceuticals and personal care products (e.g., antibiotics, psychotropic drugs, and cosmetics)</li><li>Plastics and microdebris</li></ul> | Loya, 2004; Todd et al., 2010; van Dam et al., 2011; Kroon et al., 2014; Wear and Thurber, 2015; Boucher and Friot, 2017; Littman et al., 2020; Tuholske et al., 2021; Wear et al., 2021 |

in agriculture (Table 1). In addition to the complexity of sources and types of pollutants, synergistic impacts and interactions occur when multiple pollutants are present at elevated levels, which can exacerbate the degradation of coastal ecosystems and harm associated organisms (Lu et al., 2018; Huang et al., 2021). Synergistic impacts and interactions also occur when pollutants are present with other stressors, such as climate change, disease, invasive species and overfishing. We focus on synergistic interactions on coral reefs, given the large body of research.

Herbivory is an important ecological process within coral reef ecosystems and can have complex and synergistic interactions with poor water quality (Table 2; Mumby et al., 2007). For example, in reefs with combined exposure to poor water quality and few herbivores, macroalgae and sediment-laden turfs can replace live coral as the dominant benthos (McField et al., 2020, 2022). Sea level rise and climate-driven ocean warming are predicted to increase the sensitivity of coral reef ecosystems to poor water quality. Land-based pollution can lower the threshold for thermal stress and increase coral sensitivity to infection, resulting

in increased bleaching (Fisher et al., 2019), coral mortality (Claar et al., 2020) and outbreaks of disease on coral reefs (Vega-Thurber et al., 2020). Corals that bleach from thermal stress also have reduced capacity to cope with sediment pollution (Bessell-Browne et al., 2017). Nutrient pollution can result in brittle corals that are less resilient to the impacts of climate change, such as sea level rise and the increased severity and frequency of cyclones (Table 2; Rice et al., 2020). Improving water quality through management of human activities within watersheds can therefore improve the resilience of corals to global impacts such as climate change.

## Water quality impacts on human health

Many of the same drivers of declines in water quality and aquatic biodiversity, such as watershed deforestation, forest fragmentation on riverbanks and poor coverage of sanitation services, are also associated with human health impacts (Table 2). Impacts to

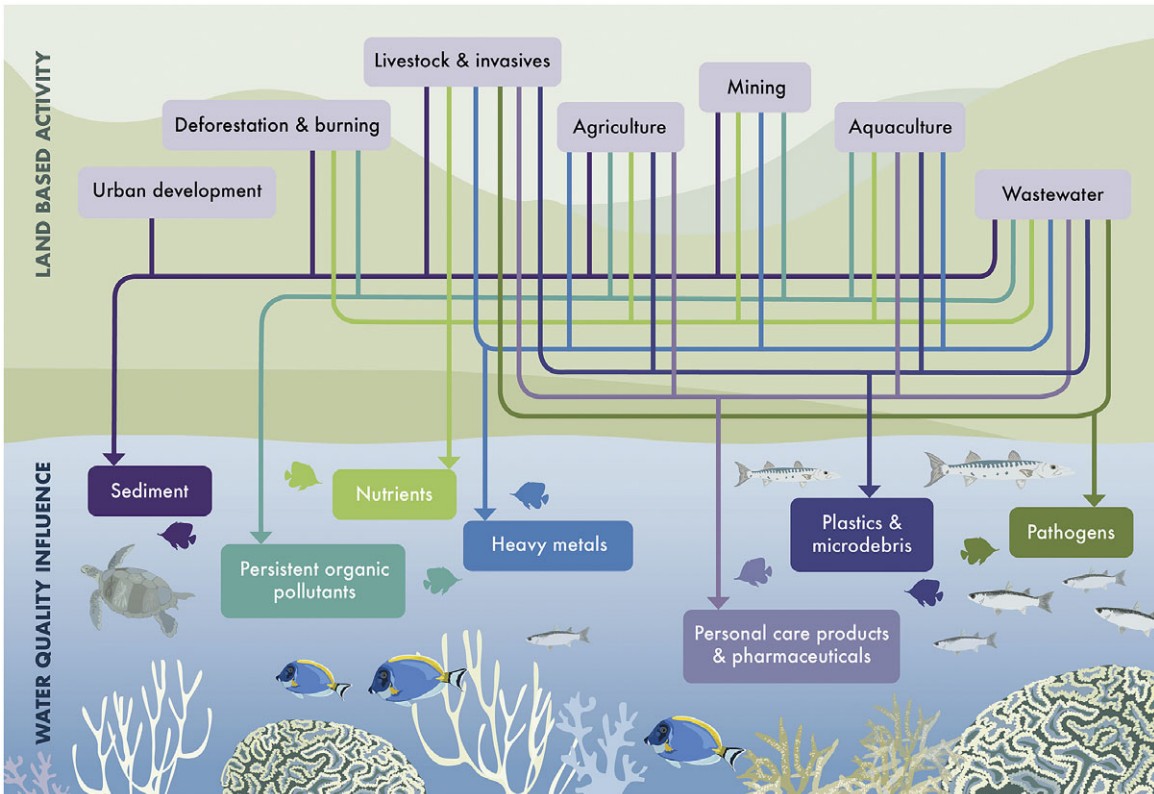

**Figure 1.** Diagram depicting flow of impacts from key land-based activities on water quality properties that reach coral reef ecosystems.

humans from poor water quality include enhanced transmission of disease through polluted water and waterways, nutrition deficits from fisheries decline and chronic illness, and food poisoning from the contamination of important aquatic foods (Shuval, 2003; World Health Organization (WHO), 2015; Chase and Ngure, 2016). Over a million people die each year from water-related diseases, and at least 50% of these deaths are children and attributable to microbial intestinal infections (Kovacs et al., 2015). Water related diseases such as diarrhoea are major contributors to global disease burdens, causing 8% of all deaths in children under the age of 5 years largely due to inadequate drinking-water quality (United Nations Inter-Agency Group for Child Mortality Estimation (UN IGME), 2019; World Health Organization (WHO) and United Nations Children's Fund (UNICEF), 2021). Persistent endemicity and explosive outbreaks of water-related disease are often fuelled by interacting environmental factors related to climate change, land use and changing social conditions (Cann et al., 2013; Prüss-Ustün et al., 2019). Water-related illness and travel associated with accessing safe water sources also contributes to reduced socioeconomic outcomes, such as reduced school attendance and gender equity (Fisher, 2008; Sorenson et al., 2011).

Communities reliant on surface and groundwater sources for drinking, bathing and household cleaning water are most at risk to water-related diseases and exposure to pollutants of emerging concern, particularly in tropical environments (Ragosta et al., 2011; World Health Organization (WHO), 2016; Herrera et al., 2017). Climate change is predicted to further increase global disease burdens by altering water-related disease dynamics (Semenza, 2020). Changes in rainfall and temperature will threaten water security, enhance pathogen survival and virulence, and increase exposure to contaminated water through multiple pathways,

including flooding (Hofstra, 2011; Levy et al., 2018). Rates of diarrhoea are predicted to increase under warmer and/or wetter conditions, with 1°C of warming predicted to increase diarrhoeal disease by 5% in developing countries (Singh et al., 2001).

Although water-related diseases are more often associated with exposure on land and freshwater, polluted seawater also presents a significant risk to human health. An estimated 180 million cases of upper respiratory disease and gastroenteritis occur each year due to humans bathing in polluted ocean waters or ingesting contaminated seafood, while around 4 million cases (and 40 thousand deaths) of infectious hepatitis A and E (HAV/HEV) occur annually from contaminated seafood from polluted coastal waters (Shuval, 2003; World Health Organization (WHO), 2015). Additionally, seafood contaminated with methylmercury and polychlorinated biphenyls can cause cardiovascular diseases in humans as well as severe impacts to infants in utero (Landrigan et al., 2020). The impacts of polluted seawater create a huge social and economic cost to communities, with pathogens in ocean pollution causing an estimated $19.4 billion (2022 USD) in economic losses annually because of their direct impacts on humans alone (Shuval, 2003).

Microplastics and debris found in wastewater pollution can also form a unique microbial community that is distinct from the surrounding water (Zettler et al., 2013). The microbial community on plastic can include pathogenic microorganisms, such as *Vibrio* spp., that cause infections through contaminated water or seafood consumption (Zettler et al., 2013; Kirstein et al., 2016). In the case of some zoonotic parasitic microbes that cause illness in aquatic wildlife and illness in humans from shellfish consumption, counts of the microbes are higher on plastics than in surrounding water (Zhang et al., 2022). Plastics therefore potentially create a novel habitat for pathogens to be concentrated and dispersed beyond

**Table 2.** Impacts of poor water quality on humans, coral reefs, and coral reef organisms categorised by pollutant type, with key references indicated for further information

| Pollutant | Impacts to health | | | |
|---|---|---|---|---|
| Terrestrially derived sediment | **Humans** | | **Human populations** | **Key references** |
| | Increased cost and complexity of water treatment. | | ► Can lead to inadequate coverage of treated water and increased time/cost accessing safe water sources. | World Health Organization (WHO), 2012, 2016; Price and Heberling, 2018; Albert et al., 2021 |
| | Increased risk of water-related diseases in humans. | | ► Increased mortality and comorbidity, healthcare burdens. | World Health Organization (WHO), 2012, 2016; Jenkins et al., 2016; Herrera et al., 2017; Albert et al., 2021 |
| | Change in aesthetic of water for human use. | | ► Loss of cultural and spiritual values, reduced tourism benefits. | World Health Organization (WHO), 2012; 2016; Landrigan et al., 2020 |
| | **Coral reef organisms** | **Coral reef ecosystems** | **Human populations** | **Key references** |
| | Reduced fertilisation and settlement for coral and reef building species. Reduced coral growth rate, colony size, and photosynthetic yield. Partial mortality. | ► Reduced reef accretion and coral cover reduces habitat complexity and the capacity of coral reef ecosystems to recover from disturbances. | ► Reduced coastal protection, tourism benefits, fisheries services, and may lead to human health impacts from reduced nutrition. | Rogers, 1990; Van Woesik and Done, 1997; Gilmour, 1999; Wesseling et al., 2001; Philipp and Fabricius, 2003; Fabricius, 2005; Bessell-Browne et al., 2017; Ricardo et al., 2018; Jones et al., 2019 |
| | Reduced coral species richness and altered coral complexity and community composition. | ► Reduced ecosystem diversity and habitat complexity, which impacts the capacity of coral reef ecosystems to recover from disturbances. | | Rogers, 1990; Edinger et al., 1998; West and van Woesik, 2001; Golbuu et al., 2008; van der Meij et al., 2010 |
| | Suppression of herbivory by reef fish. Reduced abundance of herbivorous fish species. Accumulation in algal turfs. | ► Proliferation of coral-inhibiting algae, reducing coral cover and the capacity of coral reef ecosystems to recover from disturbances. Reduced structure and nutrition also leads to reduced fish populations and diversity. | | Wenger et al., 2015; Moustaka et al., 2018; Tebbett and Bellwood, 2019; Wenger et al., 2020 |
| | Extended larval development and reduced settlement of fish. Gill damage and mortality of fish. Increased susceptibility to disease of larval fish. Reduced foraging ability. Reduced fish species richness. | ► Reduced fish recruitment, biomass and diversity, which impacts capacity of reef ecosystems to recover from disturbance. | ► Reduced tourism benefits, fisheries services, and may lead to human health impacts from reduced nutrition. | Hess et al., 2015; Wenger et al., 2015; Moustaka et al., 2018 |
| Nutrients (organic and inorganic)* | **Humans** | | **Human populations** | **Key references** |
| | Severe health impacts for human infants through consumption of contaminated water. | | ► Increased mortality and comorbidity. | World Health Organization (WHO), 2016 |
| | **Coral reef organisms** | **Coral reef ecosystems** | **Human populations** | **Key references** |
| | Reduced fertilisation, settlement and reproductive success of corals. Reduced coral growth rate Partial or complete mortality of corals | ► Reduced reef accretion and coral cover reduces habitat complexity and the capacity of coral reef ecosystems to recover from disturbances. | ► Reduced coastal protection, fisheries services, and may lead to health impacts from reduced nutrition. | Harrison and Ward, 2001; Koop et al., 2001; Cox and Ward, 2002; Loya, 2004; Todd et al., 2010; Weber et al., 2012 |
| | Reduced calcification and coral skeletal density. | ► Brittle corals that are more susceptible to breaking and erosion. This reduces the | ► Reduced coastal protection. | Edinger et al., 2000; Koop et al., 2001; Fabricius, 2005; Le Grand and Fabricius, 2011; Rice et al., 2020 |

(*Continued*)

**Table 2.**  (*Continued*)

| Pollutant | Impacts to health | | | |
|---|---|---|---|---|
| | Increased macrobioeroder density in corals. | capacity to recover from disturbances. | | |
| | Increased algal growth. | ► Proliferation of coral-inhibiting algae under reduced herbivory, reducing coral cover and the capacity of coral reef ecosystems to recover from disturbances. | ► Reduced coastal protection, fisheries services, and may lead to health impacts from reduced nutrition | McManus et al., 2000; Lapointe et al., 2011 |
| | Coral disease. | ► Potential reductions in the composition, abundance, and ultimately the accretion of coral. Limited information at present. | ► Reduced coastal protection. | Redding et al., 2013 |
| Pathogens | **Humans** | | **Human populations** | **Key references** |
| | Increased risk of water-related diseases in humans. | | ► Increased mortality and comorbidity, healthcare burdens. | Fleming et al., 2006; Sindermann, 2006; Lau et al., 2010; World Health Organization (WHO), 2016, 2019; Lamb et al., 2017 |
| | **Coral reef organisms** | **Coral reef ecosystems** | **Human populations** | **Key references** |
| | Coral disease. | ► Potential reductions in the composition, abundance, and ultimately the accretion of coral. Potential for corals to act as vectors or reservoirs of human pathogens. Limited information at present. | ► Reduced coastal protection services. Potential increase in health burdens from waterborne infectious disease. | Sutherland et al., 2011 |
| | Increased pathogenic microbiota on fish gills and shellfish. | ► Potential outbreak of disease and reductions in fish recruitment. Limited information at present. | ► Potentially reduced fisheries services, and may lead to health impacts from reduced nutrition and increase in food poisoning. | Shuval, 2003; Hess et al., 2015; World Health Organization (WHO), 2015 |
| Persistent organic pollutants (POPs) | **Humans** | | **Human populations** | **Key references** |
| | Severe health impacts (e.g., cardiovascular disease, toxicity and developmental defects) through consumption of contaminated water. | | ► Increased mortality and comorbidity, reduced schooling attendance, healthcare burdens. | World Health Organization (WHO), 2016; Landrigan et al., 2020; Müller et al., 2020 |
| | Promotion of antifungal resistant pathogens. | | ► Limited information at present. | World Health Organization (WHO), 2014; O'Neill, 2016; Woolhouse et al., 2016 |
| | Potential health impacts from immune and endocrine disruption. | | | World Health Organization (WHO), 2016 |
| | Offensive odour in water. | | ► Loss of cultural and spiritual values, reduced tourism benefits. | World Health Organization (WHO), 2012, 2016; |
| | **Coral reef organisms** | **Coral reef ecosystems** | **Human populations** | **Key references** |
| | Reduced fertilisation, settlement, and development of corals and coral reef organisms. Accumulation in coral and coral reef organisms. Reduced photosynthetic efficiency, chlorophyll concentration, and symbiont density in corals. Reduced growth of coral reef building organisms. Partial and complete | ► Reduced coral cover and reef accretion. Ultimately reduces coral reef ecosystem capacity to recover from disturbances. | ► Reduced coastal protection, fisheries services, and may lead to health impacts from reduced nutrition. | Todd et al., 2010; Turner and Renegar, 2017; Ranjbar Jafarabadi et al., 2018; Kroon et al., 2020; Nalley et al., 2021 |

**Table 2.** (*Continued*)

| Pollutant | Impacts to health | | | |
|---|---|---|---|---|
| | mortality of coral and coral reef organisms. Coral bleaching. | | | |
| | Olfactory impairment in fish.<br><br>Endocrine disruption in fish and other coral reef organisms.<br><br>Immuno-suppression in fish. | ► Limited information on chronic exposure at present. Could alter fish population dynamics and leave coral reef organism's vulnerable to additional stressors. | ► Potentially reduced fisheries services which may lead to health impacts from reduced nutrition. | Wenger et al., 2015 |
| | Accumulation in fish and molluscs. | | ► Severe disease and impacts to developing infants through consumption of contaminated seafood. | Landrigan et al., 2020 |
| Heavy metals | **Humans** | | **Human populations** | **Key references** |
| | Severe health impacts (e.g., developmental defects and toxicity) through contact with or consumption of contaminated water. | | ► Increased mortality and comorbidity, reduced schooling attendance, healthcare burdens. | World Health Organization (WHO), 2016; Rehman et al., 2018; Landrigan et al., 2020 |
| | Inhibition of biological sewage treatment. | | ► Can lead to inadequate coverage of treated water and increased time/cost accessing safe water sources. | World Health Organization (WHO), 2016 |
| | **Coral reef organisms** | **Coral reef ecosystems** | **Human populations** | **Key references** |
| | Reduced fertilisation, settlement, and development of corals. Coral bleaching. Reduced chlorophyll concentration and symbiont density in corals. Partial and complete mortality of corals. | ► Reduced coral cover and reef accretion. Ultimately reduces coral reef ecosystem capacity to recover from disturbances. | ► Reduced coastal protection, fisheries services, and may lead to health impacts from reduced nutrition. | Negri et al., 2002; Nalley et al., 2021 |
| | Embryo malformation and reduced hatching success in fish. Olfactory impairment and behavioural changes in fish. | ► Fish larvae and new recruits are potentially more prone to predation. Limited information on chronic and lower levels of exposure at present. | ► Reduced fisheries services and may lead to health impacts from reduced nutrition. Various severe health impacts from seafood consumption | Wenger et al., 2015 |
| | Immuno-suppression in fish. Accumulation in fish and molluscs.<br><br>Endocrine disruption of fish reproduction. | ► Increased disease and death rates in fish. | | Bosch et al., 2016; Landrigan et al., 2020 |
| Personal care products and pharmaceuticals | **Humans** | | **Human populations** | **Key references** |
| | Promotion of antimicrobial resistant water related pathogens. | | ► Increased mortality and comorbidity, healthcare burdens. | World Health Organization (WHO), 2014; Woolhouse et al., 2016 |
| | Potential endocrine disruption of human development and immune systems. | | ► Limited information at present. | World Health Organization (WHO), 2016 |
| | **Coral reef organisms** | **Coral reef ecosystems** | **Human populations** | **Key references** |
| | Endocrine disruption of coral fecundity. Endocrine disruption of development and/or growth of coral and coral reef organisms. | ► May lead to reductions in the composition, abundance, and ultimately the accretion of coral. Limited information at present | ► Potentially reduced coastal protection and fisheries services. | Tarrant et al., 2004; Wear and Thurber, 2015; Downs et al., 2016; Watkins and Sallach, 2021 |

**Table 2.** (*Continued*)

| Pollutant | | Impacts to health | | |
|---|---|---|---|---|
| | Reduced tissue regeneration in coral Mortality of coral Coral bleaching | | | |
| | Endocrine disruption of development and/or growth in fish Altered predator–prey interactions and aggressive behaviour of fish. | ► May lead to changes in fish population dynamics and communities | ► Potentially reduced fisheries services. | Wenger et al., 2015 |
| | DNA alterations and reduced reproduction and development in crustaceans. | ► May lead to changes in population dynamics of commercially harvested species | ► Potentially reduced fisheries services. | Garcia et al., 2014; Maranho et al., 2014 |
| | Growth inhibition in algae. | ► May lead to changes in fish population dynamics and communities, potentially causing trophic cascades in reefs | ► Potentially reduced fisheries services. | Aguirre-Martínez et al., 2015 |
| Plastic and microdebris | **Humans** | | **Human populations** | **Key references** |
| | Potential impacts to health (e.g., infertility and damage to the nervous system). Increase favourable conditions for pathogens. | | ► Limited information at present. Potential increases in disease burden associated with contaminated water. Plastic consumption potentially leading to health hazards. | Zettler et al., 2013; Galloway, 2015; Kirstein et al., 2016 |
| | **Coral reef organisms** | **Coral reef ecosystems** | **Human populations** | **Key references** |
| | Reduced reproduction and growth Disease, bleaching and tissue necrosis | ► Limited information at present. May lead to reductions in the composition, abundance, and ultimately the accretion of coral. | ► Limited information at present. Potentially reduced coastal protection services and reduced fisheries services. | Todd et al., 2010; Lamb et al., 2018; Huang et al., 2021 |
| | Enhanced transport of other contaminants to corals. | ► Limited information at present. May lead to a range of contaminant-specific impacts, such as reduced coral cover and reef accretion from sediment. | | Littman et al., 2020 |
| | Accumulation in fish, molluscs, and other reef associated organisms. Enhanced transport of pathogens and other contaminants to fish, molluscs, and other reef associated organisms. | ► Limited information at present. May lead to changes in population dynamics and communities in marine ecosystems. | ► Limited information at present. Potential increases in disease burden associated with contaminated seafood consumption. Plastic consumption potentially leading to health hazards. | Smith et al., 2018; Littman et al., 2020; Wu and Seebacher, 2020; Zhang et al., 2022 |

*Contaminant dynamics are complex, with different impacts and response curves observed even between contaminants in the same group (e.g., different heavy metals generate different impacts, different types of nutrients generate different impacts). Different levels of exposure also generate different responses, with some nutrient species generating positive responses under certain exposure levels. Impacts reported here are a general summary of known impacts from the introduction of each contaminant group at harmful levels observed in the environment.

their typical range, as floating plastics can travel longer distances than natural substrates (e.g., wood and macroalgae), and sinking microplastics are readily ingested by filter-feeding shellfish (Zettler et al., 2013; Littman et al., 2020; Zhang et al., 2022).

Polluted coastal ecosystems also affect the health of coastal human populations through fisheries decline (Hicks et al., 2019; Li et al., 2019). Millions of people depend on tropical coastal fisheries for essential protein and micro-nutrients (Kawarazuka and Béné, 2010; Teh et al., 2013). More than 10% of the global population is likely to face micronutrient and fatty acid deficiencies

if the current trajectories of fisheries decline continue, especially in the developing nations at the Equator (Golden et al., 2016). In addition, individuals already experiencing chronic health effects due to repeated exposure to pathogens will have nutrient absorption challenges, further exacerbating any micronutrient deficiencies from declining fisheries (Chase and Ngure, 2016). Better recognition of the economic and human health costs resulting from pollution impacts is critical for prioritising action and leveraging the necessary cross-sectoral partnerships and resources required for managing pollution at appropriate scales.

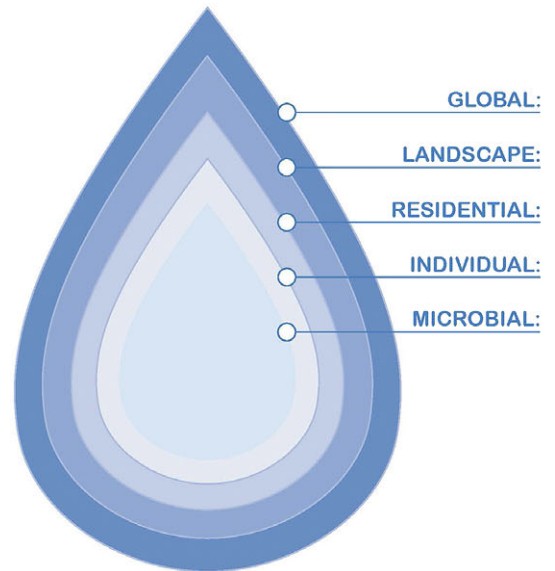

**GLOBAL:**

**Impact:** Climate-induced extreme weather events

**Intervention:** Climate negotiations to reduce greenhouse gas emissions

**LANDSCAPE:**

**Impact:** High sediment, nutrient, and contaminant loads in waterways

**Intervention:** Nature-based solutions, e.g., forest protection and soil conservation

**RESIDENTIAL:**

**Impact:** High sediment, nutrient, and contaminant loads in local water sources

**Intervention:** WASH improvements, fencing agriculture, waste management

**INDIVIDUAL:**

**Impact:** Exposure to contaminants and pathogens

**Intervention:** Handwashing, covering/treating drinking water

**MICROBIAL:**

**Impact:** Persistence of pathogens in the environment and in people

**Intervention:** Boosting immunity through vaccinations, supporting good nutrition

**Figure 2.** Nested scales of watershed processes.

## Systems approaches to watershed management

There are an array of site-based management interventions that can be implemented at nested scales within watersheds to improve water quality (Liu et al., 2017; Richmond et al., 2019; Leder et al., 2021). Mitigation efforts typically include policy instruments and place-based interventions.

Policy instruments, such as regulations or market-based incentives, can be applied at any scale and are not necessarily spatially bound within watersheds or aimed at specific watersheds. For example, the implementation of policy instruments can control, reduce and/or prevent pollution through improved use, transport, storage and disposal of chemicals (Taylor et al., 2012; Olmstead and Zheng, 2021) and nutrients (UNEP, 2012). Policy instruments can also initiate the implementation of soil conservation and erosion/runoff control strategies, such as maintaining riparian buffer zones by legislating mangrove protection (Richmond et al., 2019).

Place-based interventions are specifically applied at a range of scales, from landscape, residential, down to individual and microbial scales (Figure 2). Traditionally, human health focused place-based interventions have been targeted at a residential and individual scale, through the application of water, sanitation and hygiene (WASH) infrastructure improvements or behaviour change campaigns (World Health Organization (WHO), 2016; World Health Organization (WHO) and United Nations Children's Fund (UNICEF), 2021). However, there is now substantial evidence that landscape scale interventions could deliver significant human health outcomes, while also protecting ecosystem health. For example, a study involving 35 developing countries found that higher upstream tree cover in watersheds was associated with a lower probability of childhood diarrhoeal disease downstream (Herrera et al., 2017). In Hawaiʻi, Ragosta et al. (2011) demonstrated that higher riparian canopy cover was associated with lower *Enterococcus* concentrations in stream water. New genomics research is beginning to reveal how more intact ecosystems, from the watershed to the individual organism scale, are more likely to carry lower pathogen loads (Hess et al., 2015;

Shore-Maggio et al., 2018; Bass et al., 2019). Coastal ecosystems also play a key role in regulating disease risk in the marine environment, with a recent study showing that when seagrass meadows are present, there are 50% fewer potentially pathogenic bacteria capable of causing disease in humans and aquatic organisms (Lamb et al., 2017). However, coastal ecosystems themselves are vulnerable to high levels of pollution (Crain et al., 2009; Wear, 2016; Turschwell et al., 2021), underscoring the importance of implementing a system-wide approach when managing watersheds.

Despite the recognition that pollution is one of the greatest threats facing coral reef ecosystems (Burke et al., 2011; Andrello et al., 2021), there are limited examples of water quality management associated with successful recovery of coral reef ecosystems, and of those, the management interventions have primarily only tackled pollution arising from point-source pollution (Birkeland et al., 2013; Reef Resilience Network, 2021). Designing and measuring the effectiveness of policy instruments for water quality management is difficult due to lack of compliance and information on contaminant thresholds and monitoring (Taylor et al., 2012; Olmstead and Zheng, 2021). Place-based interventions are often impeded by difficulties in engaging stakeholders, lack of systematic/transparent planning, and funding shortfalls (Jupiter et al., 2017; Ayala-Orozco et al., 2018). For example, where stakeholders are not effectively engaged, interventions can be hindered by divergent visions, interests, and tensions within and between sectors (Ayala-Orozco et al., 2018). Lack of engagement can also limit buy-in and uptake of interventions by groups (Oteros-Rozas et al., 2015; Mitchell et al., 2022). Lack of systematic/transparent planning and evaluation can generate a lack of trust, accountability and credibility from the perspective of stakeholders (Ayala-Orozco et al., 2018), and lead to missed opportunities for effective action (Jupiter et al., 2017; Beer et al., 2020). Funding shortfalls and lack of personnel prohibit action at the scale and duration required (Ayala-Orozco et al., 2018; Beer et al., 2020). Interventions for nonpoint source pollution can be particularly challenging as pollution loading is difficult to estimate and is often attributable to many

stakeholders and sectors beholden to different regulations (Shortle and Horan, 2001).

Kāneʻohe Bay in Hawaii is a commonly cited case-study of point-source pollution (sewage) management for coral reef ecosystems resulting in a rarely seen recovery from an algal dominated back to a coral dominated state (Bahr et al., 2015). More recent successes include recovery of coral reef ecosystems within Faga'alu Bay in American Samoa and Molokaʻi in Hawaiʻi, where harmful runoff from the upstream quarry activities (Samoa) and invasive ungulate species (Hawaiʻi) were managed through targeted watershed interventions (Vargas-Ángel and Huntington, 2020). Both regions' intervention strategies required large and costly monitoring efforts to observe success, and both observed setbacks in recovery trajectories due to external disturbances (e.g., storm waves and bleaching; Bahr et al., 2015; Vargas-Ángel and Huntington, 2020).

### Watershed case study 1: Watershed interventions for systems health in Fiji

Low coverage of properly treated drinking water and sanitation in remote areas of Fiji leaves communities heavily reliant on the safety and security of unprotected water sources and vulnerable to water-related diseases. Severe outbreaks of water-related infectious diseases, such as leptospirosis, typhoid and dengue (hereafter LTD), are common. LTD cases and associated syndromes are correlated with environmental conditions, with large outbreaks typically occurring following heavy rainfall and flooding (Lau et al., 2010; Nelson et al., 2022), with increased severity within degraded watersheds (Jenkins et al., 2016).

Coastal and freshwater ecosystems are also threatened by degraded watersheds in Fiji, with decreased fish, coral and seagrass cover seen downstream of cleared and developed watersheds due to the runoff of harmful pollutants (Jenkins et al., 2010; Brown et al., 2017; McKenzie and Yoshida, 2020). These ecosystems support the livelihoods, nutrition and incomes of many rural communities (Mangubhai et al., 2018).

The Watershed Interventions for Systems Health in Fiji (WISH Fiji) project aims to address these overlapping problems through a collaborative effort between government, academic and non-governmental organisations (NGO) partners. Project collaborators are co-designing targeted 'up-stream' interventions implemented across various nested scales (Figure 2) with local communities to prevent, detect and respond to LTDs, in addition to mitigating degradation of downstream resources and ecosystems (McFarlane et al., 2019). In doing so, the WISH Fiji project aims to transform both environmental and public health action from reactive to preventative, and improve the overall health of the system to maintain integrity against LTD and natural disasters.

### Watershed case study 2: Wastewater management in Roatan, Honduras

Roatan Island, in the Bay Islands of Honduras, is bordered by coral reef ecosystems that attract over a million tourists into the region. Provisioning unpolluted runoff from watersheds is essential to maintaining the health of these ecosystems, but also to protect the health of Roatan communities and tourists. However, limited wastewater treatment on the island resulted in discharge of untreated or inadequately treated wastewater directly onto coral reef ecosystems. Local ecological knowledge linked this wastewater

runoff to outbreaks of water-related infectious disease in both humans and corals in the region, which raised fears of impacts on tourism (the main source of income in Roatan).

To combat both the human health and ecosystem impacts of untreated wastewater discharge, a collaboration between government, conservation groups and water associations identified the need for a community wastewater treatment plant (WWTP) and water quality program in West End, Roatan. The West End WWTP was then built in 2011 and has since been connected to 99% of accessible homes and businesses in the area.

Critically, a water quality laboratory led by the Bay Islands Conservation Association was also built to enable testing of marine water downstream of the WWTP, allowing significant improvements in water quality to be observed. Within 7 years of the WWTP installation, the public beach downstream passed the United States EPA safe swimming standards for *Enterococcus*, a bacteria which can cause a variety of infections and is associated with faecal contamination. The beach has since been awarded an Ecological Blue Flag certification that validates the areas as safe for tourists. Improved metrics for coral reef ecosystem health were also observed, likely as a result of improved water quality (Coral Reef Alliance, 2020).

### Key enabling factors

#### Cross-sectoral coordination and integrated governance

Managing watersheds offers numerous opportunities to address systems health challenges linked to achievement of multiple SDGs (Jenkins et al., 2018a), but simultaneously tackling multiple objectives requires coordination and integrated governance. Cross-sectoral collaborations can create a more holistic understanding of the watershed system and the breadth of its impacts across sectors (Parkes et al., 2010). This holistic understanding can improve the efficiency of integrated watershed management (IWM) by targeting multiple problems at once, creating the potential for win–win scenarios for both coastal ecosystem health and human health (Jupiter et al., 2014; Jenkins and Jupiter, 2015).

The success of cross-sectoral coordination and governance relies on careful participatory engagement and integrated policy development and implementation (Olsen and Christie, 2000; Lane, 2008). Decision-making should be developed through engagement with a wide range of stakeholders and resource users at multiple scales, improving coordination between divisions that may typically focus on the coastline or in specific sectors (Wang et al., 2016). Care should be taken to incorporate information from multiple knowledge systems in planning and practice to ensure alignment with local values and objectives (Tengö et al., 2014). Engagement should capture the diversity of land and water use practices, needs, goals and potential conflicts across sectors, and ensure that all involvement is participatory, transparent, accountable and culturally appropriate (Jupiter et al., 2014; Richmond et al., 2019).

Managing watersheds for systems health often requires coordination across multiple jurisdictions and administrative units that operate within and beyond watershed boundaries. Watershed governance is thus complicated by the mismatched boundaries of biophysical processes operating within drainage basins and jurisdictional boundaries of administrative systems responsible for land use policy implementation and health systems surveillance and

delivery (Davidson and De Loë, 2014). Polycentric and collaborative governance approaches, particularly those involving Indigenous peoples and local communities, are appropriate in this context to bridge across sectors and jurisdictional levels and address watershed systems issues at appropriate scales (e.g., Huitema et al., 2009; Morrison, 2017). Watershed management across multiple agencies and organisations can be coordinated by specific institutions that can serve as bridging organisations, such as catchment authorities, which operate most effectively when they have legislated mandates and operating budgets (Parkes et al., 2010; Davidson and De Loë, 2014).

Critically, integrated policy needs to be developed based on a good understanding of the connections among systems so that evidence-based predictions and decisions can be made about how any interventions may influence outcomes in multiple sectors (Álvarez-Romero et al., 2015). It is essential to consider any potential trade-off scenarios wherein mutual benefits are not shared between sectors, or one sector may even be exposed to more harm. For example, the construction or restoration of wetlands for improving water quality and ecosystem health may have unintended consequences for mosquito-borne disease risk (Malan et al., 2009; Horwitz and Finlayson, 2011); and the installation of dams and weirs for improving water security and sediment pollution may have unintended consequences for freshwater ecosystems and fisheries (Dudgeon et al., 2006; Kroon et al., 2014). Having a wide range of informed stakeholders sharing resources and taking an integrated approach will assist in buffering this risk and create more effective and proactive governance wherein benefits across sectors are optimised.

### Sustainable financing

Improving water quality through upstream interventions is expensive and requires sustained investment (Muchapondwa et al., 2018). There is often a long lag time between implementing interventions and observing improvements in metrics of ecosystem and public health, while success can also be obscured by other disturbances, such as cyclones and coral bleaching (Richmond et al., 2019). Delays in realising anticipated benefits create disincentives for long-term action when program and policy targets require short-term results.

Water and watershed funds are a common financing tool used in various geographies globally to ensure a sustained source of funding (The Nature Conservancy (TNC) and Goldman, 2009; Kauffman, 2014). These funds are often resourced through voluntary contributions of donors and water users, such as utility companies and farmers, which are then used to pay for and support upstream strategies to conserve the quality and security of water sources. Boards may invest the funding directly or use grants to identify and develop critical intervention strategies (The Nature Conservancy (TNC) and Goldman, 2009). Linking the needs of downstream water users with upstream communities and land users allows the funds to provide a low-cost and sustainable financing method of maintaining clean and regular water supply (The Nature Conservancy (TNC) and Goldman, 2009).

Examples of successful water funds are mainly from temperate regions and exclude marine ecosystems, such as the Latin American Water Funds Partnership (LAWFP). LAWFP is an agreement between a consortium of international NGOs to enhance and preserve water security in Latin America and supports 25 water funds across nine countries with varying water management goals and local funding bodies (Bremer et al., 2016). In total, LAWFP supported water funds are managing over 227,000 ha of land, potentially benefiting 89 million people, and have leveraged over $205 million USD in resources. Many funds prioritise not only water infrastructure management for humans, but also the use of nature-based solutions as a means to preserve the health of aquatic ecosystems (Kauffman, 2014). However, as with many water funds (and conservation efforts), there have been limited measurements of the outcomes or baselines to fully perceive the benefits of these funds (Bremer et al., 2016).

The availability of local sources of funding for sustainable financing of a water or watershed fund will vary from region to region as beneficiaries vary. Not all communities and industries pay for water use: under these circumstances, it may be feasible to develop business cases for investment based on foregone healthcare and productivity costs if watershed improvements prevent people from getting sick. Key to developing these business plans is first assessing how much disease risk can be reduced by a portfolio of management interventions and balancing the wide range of savings in foregone costs (healthcare, missed work and education, tourism impacts) against annual investment needs. Considerations also need to be taken for the potential benefits from buffering against the influence of climate change on disease.

Various other types of conservation and climate change financing can additionally or alternatively support watershed management financing. For example, in some coral reef areas, payment for ecosystem services schemes have also been proposed as a way for downstream resource users to incentivise upstream resources users to manage water quality (Goldman-Benner et al., 2012; Peng and Oleson, 2017). Climate financing that supports nature-based solutions is commonly expected to deliver various water services, though evidence shows mixed results on base flow, annual surface runoff and water quality depending on local geographic conditions and the mix of interventions utilised (Vigerstol et al., 2021; de Freitas et al., 2022).

### Conclusions/recommendations

The latest science makes it clear that unplanned development, poor land use, unsustainable agricultural practices and poor wastewater management within watersheds are significant threats to coastal populations and ecosystems. Despite the threats, incentivising improved watershed management practices for the sake of improving water quality for downstream environmental benefits has remained a challenge. In the future, it is recommended that policies and management are designed using systems health approaches that aim to restore water quality to achieve multiple benefits for human and coastal ecosystem health, while facilitating sustainable social and economic development.

Through our review, we identified a series of actionable recommendations to promote holistic approaches to watershed management for systems health (Table 3). These include best practice lessons from existing, IWM programs on: inclusive planning; implementation through cross-sectoral coordination; participatory management; monitoring to identify risks and measure progress of interventions; mobilising resources to sustain long-term action; and sharing information to promote replication and scaling of integrated approaches. To achieve SDG targets by 2030, there is increasing urgency to prioritise these types of management approaches that simultaneously deliver on benefits for nature, people and climate.

**Table 3.** Recommendations for planning, coordinating, monitoring, resourcing and scaling sustained investment in integrated watershed management for systems health

| | |
|---|---|
| Planning | Ensure engagement of the full range of actors, landowners and beneficiaries within watershed boundaries and provide platforms for transparent, participatory planning and decision-making. |
| Coordinating | Undertake policy gap analysis and strategic environmental assessments to improve harmonisation and implementation of existing policies |
| | Engage, strengthen and/or establish multi-sector management authorities (e.g., watershed commissions) with the mandate and resources to coordinate action across marine resource users/managers, logging, mining, agricultural, public health, tourism and WASH sectors. |
| Monitoring | Undertake risk and resilience assessments to identify main sources of land-based impacts to coastal reef ecosystems, and consider where these risks overlap with risks to public health, especially in the context of future climate change scenarios. |
| | Conduct research and synthesis to improve the quantity and quality of data available on thresholds and indicators of water quality and impacts on coral reef ecosystems, including integration of indigenous knowledge, citizen science, and private sector, and making the information easily accessible (i.e., through new knowledge management products and an open-source water quality database) to support monitoring and assessment programs. |
| Resource mobilisation | Develop/enhance sustainable and innovative financing mechanisms, through impact investment and private sector engagement, business case studies and integrated resource mobilisation strategies, to provide the resources required to implement phased, integrated watershed management interventions across nested scales. |
| Scaling | Develop guidance materials to integrate coral reef ecosystem health into integrated watershed management, public health and WASH planning. |
| | Document the process of developing and implementing integrated watershed management strategies in order to create communication materials for the broader conservation, WASH and public health communities on best practises and lessons learned, working with regional agencies and mechanisms to upscale. |

**Open peer review.** To view the open peer review materials for this article, please visit http://doi.org/10.1017/cft.2023.15.

**Data availability statement.** No data presented in this review.

**Acknowledgements.** Sarah Markes provided graphic design for Figure 1 and Haley Williams provided graphic design for Figure 2.

**Author contribution.** S.D.J. created the ideas and outline for the manuscript. All authors reviewed outline and provided feedback on the format and scientific content of the outline. A.Wa., S.J., A.We. and A.J. created the first written draft of the manuscript. A.Wa. edited the figures and created the tables. J.L., C.D.K., D.C., C.C., K.F., A.R. and H.S.G. provided critical revision of the scientific content of the manuscript and provided stylistic/grammatical revisions, over multiple drafts. A.Wa. addressed revisions and compiled the final manuscript.

**Financial support.** Manuscript development was supported by grant #53006 from Bloomberg Philanthropies to the Wildlife Conservation Society under the Vibrant Oceans Initiative.

**Competing interest.** No competing interest to declare.

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
