## [Reviewer Report]

*Comments to Author*: The manuscript represent a review article compiling the literature regarding watershed management and synergic effects on coastal waters and coral reefs. High quality figures and summarizing tables are very useful for readers. Minor suggestions are as follow.

Figure 1 and Table 1 present the same information in different ways. Table 1 brings a reference list although Figure 1 is very beautiful in synthetizing information. Even keeping both, they should be better explored on text. On Figure 1, merge heavy metals with persistent organic pollutants in the same box. Consider change “Personal care products & pharmaceuticals” to “Personal care, cosmetic and pharmaceutical products”.

Paragraph starting on line 122. Consider change paragraph beginning to “Pour water quality have synergic effects with herbivory…”. On line 129, revise citation Vegas-Thurben.

Table 2. Blue/green colors are will become very similar for black/white printing.

Lines 170-173. Association among microplastic and pathogens should be explained in details as a new and not obvious relationship (I did not followed related references). It seems to me that the relationship could be simple correlation (more polluted water have more pathogens) instead some kind of synergic effect (microplastic could adherirse to microbial and protect them from UV, enhancing survival). Consider present the nature of the relationship as suggested by the references.

Line 217. Change Hawai ‘i to Hawaii. The same for line 244. Although Hawai’i seams correct to native language, it is not usual for regular English.

Line 230. Correct 2020)).

Lines 236 to 238. There is something missing here.

Line 248. Two opening parenthesis.

Line 251. Is WISH “Water Innovation and Sustainability Hub”. Please, define properly.

Line 268. Define NGO (Non-Governmental Organizations).

Lines 399 to 403. Consider breaking the sentence in two.

---

## [Editor Report]

*Comments to Author*: The authors have undertaken a thorough review of all comments received and clearly reported on the changes made. The paper can now be accepted for publication. The question of Hawai’i v. Hawaii will be a decision for the publisher.